# ESTIMATION OF CROSS-LINGUAL NEWS SIMILARITIES USING TEXT-MINING METHODS

## ABSTRACT

Every second, innumerable text data, including all kinds news, reports, messages, reviews, comments, and twits have been generated on the Internet, which is written not only in English but also in other languages such as Chinese, Japanese, French and so on. Not only SNS sites but also worldwide news agency such as Thomson Reuters News [1] provide news reported in more than 20 languages, reflecting the significance of the multilingual information. In this research, by taking advantage of multi-lingual text resources provided by the Thomson Reuters News, we developed a bidirectional LSTM based method to calculate cross-lingual semantic text similarity for long text and short text respectively. Thus, users could understand the situation comprehensively, by investigating similar and related cross-lingual articles, when there an important news comes in.

## 1 INTRODUCTION

Text similarity, as its name suggests, refers to how similar the given text query is similar to the others, where we normally tend to consider mainly on their semantic characteristics, that is, how close (i.e. similar) their meanings are. Here, the text could be in the form of character level, word level, sentence level, paragraph level, or even longer, document level. In this paper, we mainly discuss the text with the form of sentences (i.e. short text) and documents (i.e. long text).

The objective of this research could be summarized in three key points. The fundamental objective is to develop algorithms for estimation of semantic similarity for the given two pieces of text written in different languages, applicable for both long text and short text, by taking advantage the untapped vast of text resources from Thomson Reuters economics news reports. Secondly, as a practical application and a verification of our model, we are aiming at development a cross-lingual recommendation system and test benchmark, where it could provide several most-related (for example, 10 results) pieces of Japanese or English text when given an English (or Japanese) article. Thirdly, we excavate cross-lingual resources from enormous the database of Thomson Reuters News and build an effective cross-lingual system by taking advantage of these un-developed treasure.

In section 2 we introduce basic concepts and prior research on semantic text similarity. It covers word level similarity, short text similarity, and long text similarity. Then we will elaborate a typical and inspiring methods for solving semantic text similarity, Siamese LSTM structure. Section 3 will explain the cores of our purposed method for cross-lingual textual similarity. Section 4 shows the detailed configurations and steps in practical experiments including the brief summary about the dataset, the methods to extract cross-lingual pairs as well as preprocessing for cross-lingual text resources. Then, we show the evaluation methods, that is, how we test our models. We first introduce two basic types of benchmark: ranks and top-N, followed by some more general criteria such as precision, recall, and F1-scores. Finally, make comparison among our purposed methods and other prior works. In section 5 we make summarization of our work and findings obtained so far and we also list some future works worth to be done.

---

[1] Official websites of Thomson Reuters: http://www.reuters.com/

## 2 RELATED WORK AND THEORIES

In spite of the length of the text, most of the state-of-the-art methods implemented based on the word embedding methods recently and thus we discuss it in detail in a separate section. To solve semantic text similarity problems, one of the most typical and inspiring methods is Siamese LSTM structure, which is considered as both basis and a competitive baseline of this research.

### 2.1 EMBEDDING TECHNIQUES FOR WORDS AND DOCUMENTS

Word embedding techniques, also known as distributed word representation, is one of the most basic concepts and application prevalent nowadays. Word embedding could be further extended to perform on even documents. The embedding techniques capturing both the semantic and syntactic information and converting them into meaningful feature vectors which help to train accurate models for natural language processing (NLP) tasks (Tang et al., 2014).

Word embedding can be implemented for both monolingual and multilingual task. There are several successful papers working on the monolingual word embedding such as the continuous bag of words models and skip-gram models (Mikolov et al., 2013), monolingual document embedding such as doc2vec (Le & Mikolov, 2014), cross-lingual word embedding (Zou et al., 2013), as well as cross-lingual document embedding model such as Bilingual Bag-of-Words without Word Alignments (BilBOWA) (Gouws et al., 2015). Through embedding model, each word, phrase or document would be converted into a fixed length vector representation, where the similarity between each of two words, phrases, or documents could be derived by calculating the cosine distance of their vector representations. Methods are distinctly different for the text data with different length when solving text similarity problem(Le & Mikolov, 2014). In respective with the length of the text, textual similarity task could be further categorized into two sub-tasks. Prevalent methods for cross-lingual document (i.e. long text) similarity could be categorized into four aspects Rupnik et al. (2016), Dictionary-based approaches(Kudo et al., 2004), Probabilistic topic model based approaches(Taghva et al., 2005), Matrix factorization based approaches(Lo et al., 2014), and Monolingual approaches.

### 2.2 TEXT SIMILARITIES USING SIAMESE LSTM

A neural network based Siamese recurrent architectures are recently proved to be one of the most effective ways for learning semantic text similarity on the sentence level. Mueller, in his work, implements a Siamese recurrent structure called Manhattan LSTM (MaLSTM) Mueller & Thyagarajan (2016), which is practically used as the estimation of relativeness (i.e. similarity) when given any two sentences in English. This structure using Long Short-Term Memory (LSTM)Hochreiter & Schmidhuber (1997) have state-of-the-art performance on both semantic relatednesses scoring task and entailment classification using the SICK database, one of the NLP challenges provided by SemEvalAgirre et al. (2016). This model could identify how two sentences are similar to each other by trying to "understand" their true meaning on a deeper aspect, like the sentence pairs "He is smart" and "A truly wise man" as the figure demonstrates. They have no common word with different length, but they are indeed highly relevant to each other in terms of their implications, which a human cannot recognize without more consideration and logical analysis, suggesting the difficulty of this challenge.

In our work, we developed a new recurrent structure inspired by MaLSTM, by modifying the Siamese (i.e. symmetric) LSTM modules to an "unbalanced" ones, and add a full-connect neural network layer following the output of LSTM modules, which is more flexible and effective over text similarity task.

## 3 A BIDIRECTIONAL LSTM BASED METHOD

We implement the two independent modules of bi-directional LSTM recurrent neural networks on both English input and Japanese input respectively and the overview of this structure is shown in the figure 1. We use the cross-lingual training data in the form of pre-trained word vectors as input. Feed the word vector sequentially to LSTM modules. This is discussed in detail in the section 2.1. Furthermore, as a limitation of our LSTM modules, we have uniform length of data as input, denoted as "maxlen". The residue of the parts of sequence longer than maxlen will be abandoned,

while those input sequence shorter than "maxlen" will be padded with a predefined value (i.e. a word) such as "null" at the tail so that all the input data could be same in length. The two bi-LSTM modules are responsible for English sequence and Japanese Sequence respectively. They generate four hidden layer outputs and we concatenate them into a joint feature. Details is elaborated in the 3.1. The joined feature is further fed into a densely-connected neural network of 1 depth, resulting in 1 dimension output $y \in [0,1]$ as the final similarity score of the two inputs cross-lingual data, by means of regression. In general, LSTM-based model pay more attention on the order information of the input sequence, which might significant determine the real meaning of the a sentence written in natural languages.

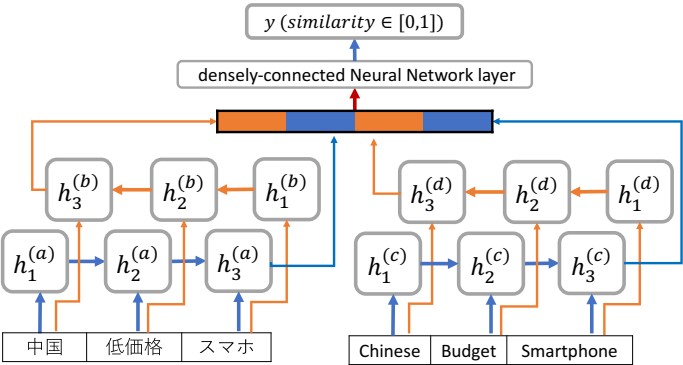

Figure 1: Illustration of LSTM-based method

## 3.1 THE BI-LSTM LAYER

In this research, we take advantage of bi-LSTM (bi-directional long short-term memory), to enhance the ordinary RNN performance considering both forward and backward information and solve the problem of the long-term dependencies. The updates rules of LSTM for each sequential input $x_1, x_2, ..., x_t, ..., x_T$ could be express as:

$$i_t = \text{sigmoid}(W_i x_t + U_i h_{t-1} + b_i) \tag{1}$$

$$f_t = \text{sigmoid}(W_f x_t + U_f h_{t-1} + b_f) \tag{2}$$

$$\tilde{c}_t = \tanh(W_c x_t + U_c h_{t-1} + b_c) \tag{3}$$

$$c_t = i_t \odot \tilde{c}_t + f_t \odot c_{t-1} \tag{4}$$

$$o_t = \text{sigmoid}(W_o x_t + U_o h_{t-1} + b_o) \tag{5}$$

$$h_t = o_t \odot \tanh(c_t) \tag{6}$$

where $h_{t-1}$ is the hidden layer value of the previous states and the sigmoid and tanh functions in the above equations are also used as activation functions:

$$\text{sigmoid}(x) = \frac{1}{1 + \exp(-x)} \tag{7}$$

$$\tanh(x) = \frac{2}{1 + \exp(-2x)} - 1 \tag{8}$$

The weights (i.e. parameters) we need to train include $W_i, W_f, W_c, W_o, U_i, U_f, U_c, U_o$ and bias vectors $b_i, b_f, b_c, b_o$. In this layer, there are four LSTM modules, constructing two bi-LSTM structures, where we only consider the final output (i.e. final value of the hidden layer) of each LSTM modules: LSTM-a read Japanese text forwardly. The value of hidden layer is denoted as $\mathbf{h}_i^{(a)}$ where

i is the i-th input of the sequence, while LSTM-b read backwardly denoted as $\mathbf{h_i^{(b)}}$. Symmetrically, LSTM-c and LSTM-d are used to read English text, denoted as $\mathbf{h_i^{(c)}}$ and $\mathbf{h_i^{(d)}}$. As the results, we obtain four feature vectors derived from hidden layer values of the four LSTM modules, keeping all necessary information regarding to the cross-lingual inputs. We then merge these four features by concatenating them directly:

$$\mathbf{x_{i,j}} = (\mathbf{h_L^{(a)}}, \mathbf{h_L^{(b)}}, \mathbf{h_L^{(c)}}, \mathbf{h_L^{(d)}}) \tag{9}$$

where i and j refer to the document number of the input text for Japanese and English respectively, and vector $\mathbf{h_L^{(a,b,c,d)}}$ refers to the final status (i.e. the value) of the hidden layers of the LSTM module after feeding the last (or the first word, if backwardly) word.

## 3.2 DENSE LAYER

We use the most basic component of basic full-dense Neural Network layer as the top layer. The function of this layer could be expressed as:

$$y_{i,j} = f(\mathbf{w}^T \mathbf{x_{i,j}} + b) \tag{10}$$

Here, the function $f$ is also known as "activation" function, and $b$ is the one dimension bias for the neural network and $\mathbf{w}$ is the weight (i.e. the parameters to be trained) of the neural network. In this project, we mainly apply the Rectified Linear UnitReLu Nair & Hinton (2010) function as activation function in the dense layer :

$$f(x) = \ln[1 + \exp(x)] \tag{11}$$

As for the optimization, although we are handling a classification problem, based on the experiment results, we find that, instead of using ordinary cross-entropy cost, it performs better if we use Quadratic cost (i.e. mean square error) as the cost function, which could be described as:

$$C = \sum_{v=1}^{N} (y_{true,v} - y_{pred,v})^2 \tag{12}$$

where N is the total number of the training data, while $y_{true,v}$ and $y_{pred,v}$ refer to the true similarity and the predicted similarity respectively. In practice, the stochastic gradient descent (SGD) is implemented by means of the back-propagation scheme. After computing the outputs and errors based on the cost function $J$, which is usually equal to the negative log of the maximum likelihood function, we update parameters by the gradient descent method, expressed as:

$$\mathbf{w} \leftarrow \mathbf{w} - \varepsilon \nabla_w J(\mathbf{w}) \tag{13}$$

where $\varepsilon$ is known as "learning rate", defining the update speed of the hyper-parameters $\mathbf{w}$. However, the training process might fail due to either improper initialization regarding weights or the improper learning rate value set. Practically, based on the results of the experiments, the best performance is achieved by applying the Adam optimizer Kingma & Ba (2014) to perform the parameter updates.

## 4 EXPERIMENTS AND RESULTS

### 4.1 EVALUATION METHODS

We use mainly two categories of the evaluations, TOP-N benchmark based on ranks, and traditional criteria for classification such as precision, recall as well as the F1-value. As the applications of this project is to suggest the users several cross-lingual (For instance, English) alternatives news when the user provides a Japanese article as a query, we make the system pick up 1, 5 and 10 of the most similar Japanese alternatives during the evaluation process. The figure 2 illustrates the relationship and evaluation procedures for ranks, TOP-N index.For a given Japanese text (i.e.the query) $J_x$, calculate the similarity score between $J_x$ and all English text of test data sets $(E_1, E_2, ..., E_x, ..., E_M)$ to derive a list of scores $L_x = (S_{x,1}, S_{x,2}, ..., S_{x,x}, ..., S_{x,M})$ , where the corner mark M is the total number of English documents to be considered, and $E_x$ is the true similar article for with similarity score of 1. Then sort this list in the order from large to small and find out the rank (i.e. position, index) of the score $S_{x,x}$ inside this sorted list noted as $R_x$, the rank for the query document $J_x$. Repeat this process recursively for N Japanese articles $(J_1, J_2, ..., J_N)$, result in a list of ranks

$R = (R_1, R_2, ..., R_N)$ regarding the collections of $J_x$. Then we take the number of query documents with ranks smaller than N as TOP-N. In other words, TOP-1 refers to the number of query documents with rank equal to 1 and TOP-5 refers to the number of a query with rank equal and smaller than 5.

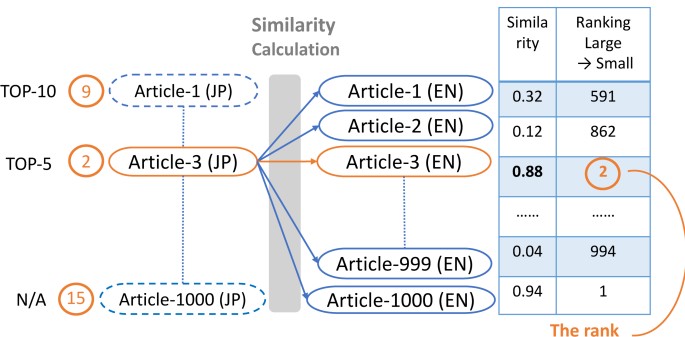

Figure 2: Illustration of an evaluation procedures using ranks and TOP-N index

## 4.2 BASELINE: SIAMESE LSTM WITH GOOGLE-TRANSLATION

As Siamese LSTM is one of the deep learning-based models with the art-of-state performance on the semantic text similarity problems. In this research, we make this model be a baseline by extending this model from monolingual domain to cross-lingual domain with the help of the Google Translation services. We first translate all Japanese text into English version on both test and training data by using the google translate service [2] . Then we implement the Siamese LSTM model as described in the original paper for Siamese LSTM Mueller & Thyagarajan (2016) with the help of the open source code on the Github [3]. The illustration of this baseline method regarding a two cross-lingual input, we first translate the Japanese input into an English sentence using Google Translation service. Then, we can consider the cross-lingual task as monolingual one using so that we can apply the Siamese LSTM model for training as a baseline.

## 4.3 DATASETS AND PRE-PROCESSING

Table 1: Example of similarity relationship for Japanese words (translated)

| | Toyota | | Sony | |
|---|---|---|---|---|
| TOP | word | Similarity | word | Similarity |
| 1 | Honda | 0.612 | PlayStation | 0.612 |
| 2 | Toyota corp | 0.546 | Entertainment | 0.546 |
| 3 | Hyundai corp | 0.536 | SonyBigChance | 0.536 |
| 4 | Chrysler | 0.524 | Game console | 0.524 |
| 5 | Nissan | 0.519 | Nexus | 0.519 |
| 6 | motor | 0.511 | X-BOX | 0.511 |
| 7 | LEXUS | 0.506 | spring | 0.506 |
| 8 | Acura | 0.493 | Windows | 0.493 |
| 9 | Mazda | 0.492 | Compatibility | 0.492 |
| 10 | Ford | 0.486 | application software | 0.486 |

Thomson Reuters news [4] is a worldwide news agency providing worldwide news in multiple languages. Most of the reports are originally written in English and translated and edited into other

---

[2]Google Translation Web API could be accessed from https://github.com/aditya1503/Siamese-LSTM

[3]The open source code for Siamese LSTM can be accessed from https://github.com/aditya1503/Siamese-LSTM

[4]Official websites of Thomson Reuters: http://www.reuters.com/

languages including Chinese, Japanese and so on. These multi-lingual texts are expected to be highly potential resources for tasks related the multi-lingual natural languages processing. In this research, we use 60,000 news articles in 2014 from Thomson Reuters News related to the economics. The preprocessing of text, we convert raw data to normalized ones, which could be further used to train word2vec models for both English and Japanese text respectively. We train Japanese word2vec model and English word2vec model separately using news articles with same contents in 2014. In our experiment, we use the model of Continuous Bag of Words (CBOW), with fixed 200 dimensions of word embedding. Other parameters are set using default value used in Gensim package [5].

Table 2: Example of similarity relationship for English words

| TOP | lexus | | lenovo | |
|---|---|---|---|---|
| | word | Similarity | word | Similarity |
| 1 | acura | 0.636 | huawei | 0.636 |
| 2 | corolla | 0.588 | zte | 0.588 |
| 3 | camry | 0.571 | xiaomi | 0.571 |
| 4 | 2002-2005 | 0.570 | dell | 0.570 |
| 5 | sentra | 0.541 | handset | 0.541 |
| 6 | prius | 0.539 | smartphone | 0.539 |
| 7 | 2003-2005 | 0.537 | hannstar | 0.537 |
| 8 | sedan | 0.533 | thinkpad | 0.533 |
| 9 | mazda | 0.530 | tcl | 0.530 |
| 10 | altima | 0.524 | medison | 0.524 |

As discussed in the section 2.1, the word2vec could build relationships among words based on their original context. We could find several most similar words when given a query word by calculating their cosine similarity. The table 2 and 1 demonstrate examples to find the most similar words when given a word query in English and in Japanese respectively. All these results suggest the effectiveness of word2vec algorithms and successful of the training processes.

## 4.4 EXPERIMENTS ON SHORT TEXT

### 4.4.1 TRAINING DATA

On Short Text, we firstly pick up 4000 pairs of parallel Japanese-English cross-lingual news titles from the database with the period from the January to February of 2014, all of which are labeled with a similarity score of 1. To provide balance training data, we also generated 4000 pairs of un-parallel Japanese-English cross-lingual news titles by a random combination. In order to simplify our model and experiments, we use the assumption that the similarity the random combination of Japanese text and English text is 0. Then on Long Text, similar to the data preparation of experiments for short text introduced, we prepare 4000 parallel (i.e. similarity = 1) Japanese-English news articles and 4000 un-parallel (i.e. similarity = 0) ones for training data through random combination.

### 4.4.2 TEST DATA

On Short Text, in order to evaluate our model more comprehensively, we have prepared two sets of independent test data. TEST-1S contains 1000 pairs of parallel Japanese-English news titles, selected and split from the same period of training data, from January 2014 to middle of February in 2014. Similar, TEST-2S contains title pairs with time stamps of December 2014. On Long Text, similar to the case of short test evaluation, we have prepared two sets of independent test data. For training data, we prepared similar dataset as short text experiments. TEST-1L and TEST-2L contain 1000 pairs of parallel Japanese-English news long articles respectively.

---

[5]To see more specific of the configuration of word2vec model, see the documentation of Word2Vec class from https://radimrehurek.com/gensim/models/word2vec.html

### 4.4.3 RANKS AND TOP-N

Figure 3 illustrates how TOP-N (N=1,5,10 with the color red, blue and green respectively) indices change along with the 10 training epochs regarding the two test data TEST-1S and TEST-2S in dash line and bold line respectively. For both test data, TEST-1S and TEST-2S, all the 3 index, TOP-1, TOP-5 and TOP=10 of them gradually increase during the training, proving the effectiveness of the training process. Table 3 shows statistics information of the results for LSTM based model on short text data. This table shows the distribution of the rank score of all test documents as well as the TOP-N scores. In addition to the mean value, standard deviation, minimum value and maximum value, we also list the value for percentiles of 25th, 50th, and 75th. Figure 4 illustrates how TOP-N (N=1,5,10 with the color red, blue and green respectively) indices change along with the 10 training epoch regarding the two test data TEST-1L and TEST-2L in dash line and bold line respectively. For both test data, TEST-1L and TEST-2L, all the 3 index, TOP-1, TOP-5 and TOP=10 of them gradually increase during the training, proving the effectiveness of the training process. Table 3 shows statistics information of the results for LSTM based model on Long text data. The performance of successful recommendation numbers from our bi-LSTM based model

Table 3: Summary of in terms of TOP-N benchmark

| TOP-10 | | | | |
|---|---|---|---|---|
| | SHORT | | LONG | |
| method | TEST-1S | TEST-2S | TEST-1L | TEST-1L |
| LSTM | **511** | **495** | **456** | **432** |
| baseline | 243 | - | 302 | - |
| TOP-5 | | | | |
| | SHORT | | LONG | |
| method | TEST-1S | TEST-2S | TEST-1L | TEST-1L |
| LSTM | **339** | **338** | **284** | **278** |
| baseline | 134 | - | 192 | - |
| TOP-1 | | | | |
| | SHORT | | LONG | |
| method | TEST-1S | TEST-2S | TEST-1L | TEST-1L |
| LSTM | **90** | **106** | **61** | **58** |
| baseline | 39 | - | 50 | - |

is twice of the baseline. When the baseline method calculating the similarity of two sentences, no matter whether there are different types of word arrangement for the two input, or there are different words used referring to the same meaning, which proves the effectiveness of encoding (i.e. embedding) ability for input text. However, the baseline model has the "Siamese LSTM structure", which means, in other words, the two LSTM instances always share the same parameters during the training. This might be effective for a monolingual case, but not good enough on the cross-lingual case. Thus, the LSTM instances used in our purposed model are all independently holding their own unique parameters. In addition, the bi-directional structure also helps to encode the feature of each input text more comprehensively. Finally, instead of using cosine similarity as the final layer in the baseline method, we used the fully connected neural network as output, making the output layer adjust (i.e. train) its parameters so as to learn precise patterns from the features generated by LSTMs. We believe these three modifications improve the final results for our LSTM-based model.

## 5 CONCLUSION

We developed a bi-LSTM-based model to calculate cross-lingual similarities given a pair of English and Japanese articles. Instead of using a translation module or a dictionary to translate from one to another language, our model has outstanding performance with short text. Furthermore, we modify and implement a popular Siamese LSTM model as the baseline and we found both of our models outperform the baseline. For practical testing, we defined the concept of "TOP-N" and "ranks" to

test the overall performance of the model, with visualized results. We also make a comparative study based on the results of the experiments that bi-LSTM based obtains better performance on short text data such as news title and alert message, which is averagely shorter than 20 words, a contrast to the normal news articles with more than 200 words in average. As the results, both models obtained satisfactory performance with over half of the test documents of 1000 holding ranks lower than 10 (i.e. TOP-10). As a high-performance cross-lingual news calculating system, we expect that it could achieve optimal performance by taking advantages of both two models as a complete system.

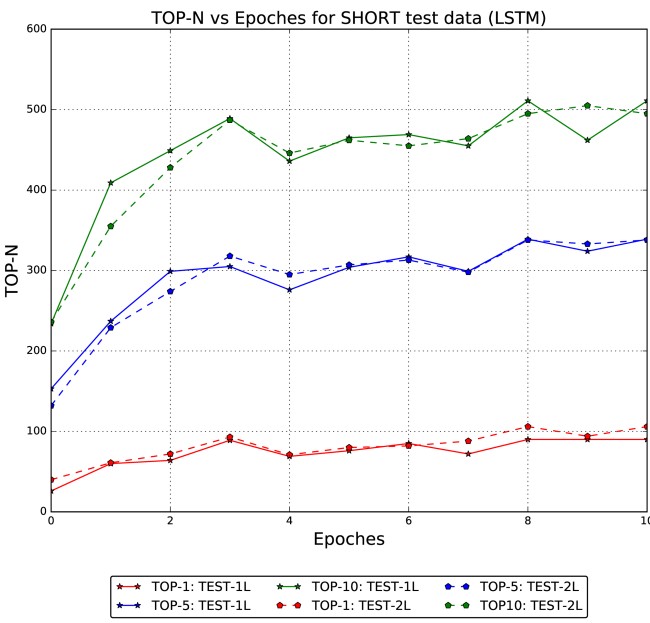

Figure 3: TOP-N along with Epoches for TEST-1S, TEST-2S, **Short** data, **LSTM**-based model

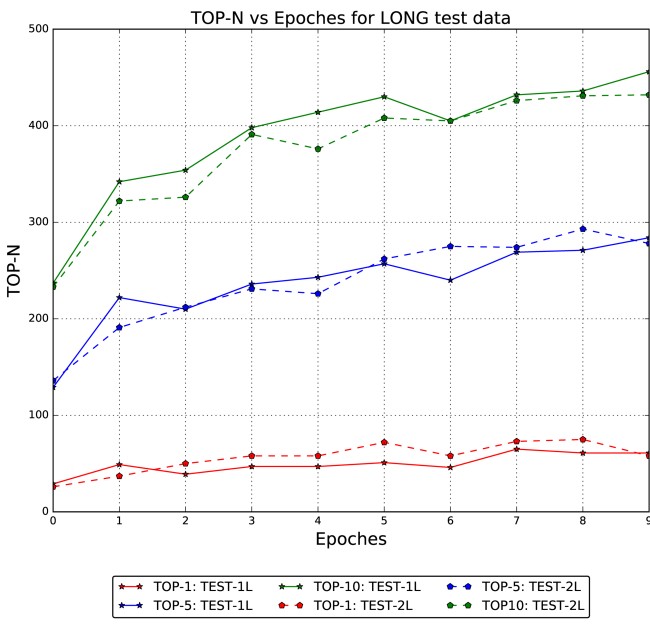

Figure 4: TOP-N along with Epoches for TEST-1L, TEST-2L, **Long** data, **LSTM**-based model

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
