# OpenReview forum: "Estimation of cross-lingual news similarities using text-mining methods"
_ICLR.cc/2018/Conference — Reject_

### Official Review · AnonReviewer1 · 2017-11-23
**This paper is hard to read and need proof-reading by a person proficient in English. The experiments are extremely limited, on a toy task. No other baseline than (Mueller and Thyagarajan, 2016) is considered. The related work section lacks important references. It is hard to find positive points that would advocate for a presentation at ICLR.**

**Rating:** 2
**Confidence:** 5

**Review:**

* PAPER SUMMARY *

This paper proposes a siamese net architecture to compare text in different languages. The proposed architecture builds upon siamese RNN by Mueller and Thyagarajan. The proposed approach is evaluated on cross lingual bitext retrieval.

* REVIEW SUMMARY *

This paper is hard to read and need proof-reading by a person proficient in English. The experiments are extremely limited, on a toy task. No other baseline than (Mueller and Thyagarajan, 2016) is considered. The related work section lacks important references. It is hard to find positive points that would advocate for a presentation at ICLR.

* DETAILED REVIEW *

On related work, the authors need to consider related work on cross lingual retrieval, multilingual document representation:

Bai, Bing, et al. "Learning to rank with (a lot of) word features." Information retrieval 13.3 (2010): 291-314. (Section 4).

Schwenk, H., Tran, K., Firat, O., & Douze, M. Learning Joint Multilingual Sentence Representations with Neural Machine Translation, ACL Workshop on Representation Learning for NLP, 2017

Karl Moritz Hermann and Phil Blunsom.  Multilingual models for compositional distributed semantics. In ACL 2014. pages 58–68.

Hieu Pham, Minh-Thang Luong, and Christopher D. Manning. Learning distributed representations for multilingual text sequences. In Workshop
on Vector Space Modeling for NLP. 2015

Xinjie Zhou, Xiaojun Wan, and Jianguo Xiao. Cross-lingual sentiment classification with bilingual document representation learning. In ACL 2016

...

On evaluation, the authors need to learn about standard retrieval evaluation metrics such as precision at top 10, etc and use them. For instance, this book will be a good read.

Baeza-Yates, Ricardo, and Berthier Ribeiro-Neto. Modern information retrieval. Vol. 463. New York: ACM press, 1999.

On learning objective, the authors might want to read about learn-to-rank objectives for information retrieval, for instance,

Liu, Tie-Yan. "Learning to rank for information retrieval." Foundations and Trends in Information Retrieval 3.3 (2009): 225-331.

Burges, Christopher JC. "From ranknet to lambdarank to lambdamart: An overview." Learning 11, no. 23-581 (2010): 81.

Chapelle, Olivier, and Yi Chang. "Yahoo! learning to rank challenge overview." Proceedings of the Learning to Rank Challenge. 2011.

Herbrich, Ralf, Thore Graepel, and Klaus Obermayer. "Large margin rank boundaries for ordinal regression." (2000).

On experimental setup, the authors want to consider a setup with more than 8k training documents. More importantly, ranking a document set of 1k documents is extremely small, toyish. For instance, (Schwenk et al 2017) search through 1.5 million sentences. (Bai, Bing, et al 2009) search through 140k documents. Since you mainly introduces 2 modifications with respect to (Mueller and Thyagarajan, 2016), i.e  (i) not sharing the parameters on both branch of the siamese and (ii) the fully connected net on top, I would suggest to measure the effect of each of them both on multilingual data and on the SICK dataset used in (Mueller and Thyagarajan, 2016).

---

### Official Review · AnonReviewer2 · 2017-11-27
**In this paper, authors propose a recurrent structure inspired by MaLSTM, by taking advantage of multilingual textual resources provided by the Thomson Reuters News to calculate cross-lingual semantic text similarity for long text and short text.**

**Rating:** 6
**Confidence:** 4

**Review:**

In the Following, pros and cons of the paper are presented.

Pros
-------

1. Many real-world applications.
2. Simple architecture and can be reproduced (if given enough details.)


Cons
----------------------

1. Ablation study showing whether bidirectional LSTM contributing to the similarity would be helpful.
2. Baseline is not strong. How about using just LSTM?
4. It is suprising to see that only concatenation with MLP is used for optimization of capturing regularities across languages.
5. Equation-11 looks like softplus function more than vanilla ReLU.
6. How are the similarity assessments made in the gold standard dataset. The cost function used only suggest binary assessments. Please refer to some SemEval tasks for cross-lingual or cross-level assessments. As binary assessments may not be a right measure to compare articles of two different lengths or languages.

Minor issues
------------

1. SNS is meant to be social networking sites?
2. In Section 2.2, it denote that 'as the figure demonstrates'. No reference to the figure.
3. In Section 3, 'discussed in detail' pointed to Section 2.1 related work section. Not clear what is discussed in detail there.
4. Reference to Google Translate API is wrong.


The paper requires more experimental analysis to judge the significance of the approach presented.

---

### Official Review · AnonReviewer3 · 2017-11-27

**Rating:** 2
**Confidence:** 4

**Review:**


The paper studies the problem of estimating cross-lingual text similarity by mining news corpora. The motivation of the problem and applications are presented well, especially for news recommender systems.

However, there are no novel scientific contributions. The idea of fusing standard bi-LSTM layers coupled with a dense fully-connected layer alone is not a substantial technical contribution. Did they try other deep architectures for the task? The authors cite some previous works to explain their choice of approach for this task. A detailed analysis of different architectures (recurrent and others) on the specifc task would have been more convincing.

Comparison against other relevant baselines (including other cross-lingual retrieval approaches) is missing. There are several existing works on learning cross-lingual word embeddings (e.g., Mikolov et al., 2013). Some of these also make available pre-trained embeddings in multiple languages. You could combine them to learn cross-lingual semantic similarities for the retrieval task. How does your approach compare to these other approaches besides the Siamese LSTM baseline?

Overall, it is unclear what the contributions are — there has been a lot of work in the NLP/IR literature on the same task, yet there is no detailed comparison against any of these relevant baselines. The technical contributions are also not novel or strong  to make the paper convincing.

---

### Decision · Program_Chairs · 2018-01-29
**ICLR 2018 Conference Acceptance Decision**

**Decision:**

Reject

**Comment:**

The pros and cons of this paper cited by the reviewers can be summarized below:

Pros:
* The motivation of the problem is presented well
* The architecture is simple and potentially applicable to real-world applications

Cons:
* The novel methodological contribution is limited to non-existant
* Comparison against other relevant baselines is missing, and the baseline is not strong
* The evaluation methodology does not follows standard practice in IR, and thus it is difficult to analyze and compare results
* Paper is hard to read and requires proofreading

Considering these pros and cons, my conclusion is that this paper is not up to the standards of ICLR.